# Community health volunteers' experiences during the COVID-19 pandemic in Kiambu county, Kenya: A qualitative study

**Naomi Wachira[1], Prabhjot Kaur Juttla[2]\*, Bernard Kimani[1], Moses Kamita[3], Samuel Mungai[3], James Ndimbii[4], Francis Makokha[3], Magoma Mwancha-Kwasa[1]**

**1** Department of Health, County Government of Kiambu, Kiambu, Kenya, **2** School of Medicine, College of Health Sciences, University of Nairobi, Nairobi, Kenya, **3** School of Public Health, Mount Kenya University, Thika, Kiambu, Kenya, **4** Elizabeth Glaser Pediatric AIDS Foundation, Nairobi, Kenya

\* pkjuttla13@students.uonbi.ac.ke

## Abstract

### Background

For already overburdened health systems in low- and middle-income countries (LMICs), the COVID-19 pandemic presented an almost impossible challenge. In Kenya, efforts to mitigate the impact of the pandemic included the mobilization of community health volunteers (CHVs), a cadre that has been historically understaffed and under-resourced. These volunteers were required to sustain the delivery of routine community-based health services while also taking on additional responsibilities related to COVID-19 mitigation. This study explored the challenges faced by CHVs during the COVID-19 response in Kiambu County; focusing on their experiences with control measures, impacts on community-level healthcare delivery, and perspectives on the government's pandemic response within the community health framework.

### Methods

This study employed a phenomenological exploratory qualitative design. Due to logistical constraints, only two focus group discussions could be conducted with 24 CHVs, representing 48% of eligible participants who met the inclusion criteria: a minimum of five years of experience, active involvement in the pandemic response, and availability for the interview. Data were digitally recorded, transcribed, translated, and coded for thematic analysis.

### Results

Specific themes from the experiences of the CHVs during the COVID-19 lockdown were: (1) dedication and commitment to serving the community; (2) overcoming demoralization; and (3) community barriers to health care delivery and access. In regards to the community perspectives of COVID-19, the CHVs relayed widespread

**Data availability statement:** All relevant data are within the paper and its Supporting Information files.

**Funding:** The author(s) received no specific funding for this work.

**Competing interests:** The authors have declared that no competing interests exist.

misinformation among community members, with experiences of stigma due to COVID-19 misconceptions. They also gave second-person accounts of the economic strife the community went through as a result of mitigation measures. The CHVs were skeptical in the county's preparedness in dealing with both the COVID-19 and future pandemics.

## Conclusion

Despite facing stigma, misinformation, limited resources, and economic hardships, CHVs demonstrated commitment to their roles. Their efforts not only underscored their resilience but also revealed critical gaps in preparedness and resource allocation within the healthcare system. However, the small number of FGDs and insufficient data saturation should be considered when interpreting the findings. Nonetheless, our study provides a starting point for further research and comparative analysis across other counties in Kenya.

## Introduction

Community health volunteers (CHVs) are recognized by the World Health Organization as "a variety of community health auxiliaries who work in their own communities" [1]. These volunteers play a critical role in advancing universal health coverage (UHC), particularly in low- and middle-income countries where access to essential healthcare services remains a significant challenge [2,3]. In these settings, UHC relies heavily on the extension of primary health care into local communities, with CHVs serving as the first point of contact for healthcare delivery: contributing to the accessibility and effectiveness of the healthcare system as a whole [4].

In Kenya, CHVs are key participants in the national community health strategy, which is designed to promote community involvement in healthcare decision-making and delivery. The strategy is managed by community health strategy coordinators (CHSCs) who oversee CHV activities and ensure that volunteers have access to the necessary tools and resources for service provision. CHVs are entrusted with the important task of promoting preventive health measures, supporting health education, and ensuring the continuity of care within their communities [5].

Historically, CHVs in Kenya operated in an informal capacity, without formal recognition or integration into the national healthcare system [6]. Consequently, the recruitment and retention of volunteers remained limited. Before the enactment of the Public Health Act in October 2023, CHVs functioned as volunteers, largely driven by local initiatives and their personal commitment to improving public health [7]. Their roles, which primarily involved delivering health information and encouraging healthcare-seeking behaviour, were *ad hoc* and lacked structure [6]. Training for CHVs, when/if provided, was inconsistent and varied across regions, without standardized guidelines to ensure uniformity in knowledge and skill. Moreover, these

volunteers often operated without compensation, relying on their dedication to improving health outcomes in underserved areas.

The outbreak of the COVID-19 pandemic highlighted gaps in the delivery of community-based healthcare services in the Kenyan healthcare system. Since the first confirmed case in March 2020 [8], the pandemic jolted healthcare systems globally, and Kenya was no exception. In response, CHVs were enlisted as key participants in Kenya's COVID-19 Response Strategy [8,9]. Their roles were expanded to include promoting behavioral changes, combating misinformation, encouraging hygienic practices, and sustaining efforts to address other priority diseases at the community level [9,10]. CHVs also contributed to contact tracing efforts and monitored vulnerabilities within their communities.

Despite these critical contributions, CHVs continued to work as volunteers throughout the pandemic, depending on the government, through CHSCs, for the provision of personal protective equipment (PPE), sanitizers, and other necessary supplies [5]. They were also expected to receive psychosocial support to help manage the increased demands of their roles. Unfortunately, these measures were often insufficient, underscoring gaps in Kenya's healthcare system and raising concerns about its preparedness and resilience during health crises.

Given the ongoing legislative reforms aimed at strengthening Kenya's healthcare system in an effort to advance UHC [11], it is crucial to examine the experiences of CHVs during the pandemic. This study seeks to explore the challenges faced by CHVs and the community members they served while implementing COVID-19 control measures. By analyzing their lived experiences and evaluating their perspectives on government crisis management at the community level, this research offers insights into the role of CHVs and contributes to the broader discourse on improving healthcare system preparedness in similar settings.

## Materials and methods

### Study design

The analysis presented in this paper adopted a phenomenological exploratory qualitative approach using focus group discussion (FGDs).

### Study setting and local context

The study was conducted in Kiambu County, located in the central region of Kenya, covering an area of 2,543.5 km². According to the 2019 census, the average household size in the county is five individuals. At the time of the study, Kiambu had 795,241 households and 132 community health units (CHUs). The CHV-to-household ratio was 1:100, and the community unit-to-household ratio was 1:1,000 [5]. The county was served by 98 health facilities distributed across various levels of care, including three Tier 5 inter-county facilities, 13 Tier 4 hospitals, 24 Tier 3 health centers, and 70 Tier 2 dispensaries.

During the COVID-19 pandemic, Kiambu adopted a centralized approach to managing COVID-19 cases, designating Tigoni Level 4 Hospital as the sole facility for COVID-19 services [12]. As a result, all COVID-19 patients were referred to this hospital during the study period. At the time of the study, Kiambu County had the second-highest number of confirmed COVID-19 cases in Kenya [13]. The county reported its first COVID-19 case in March 2020, and by August 2022, the total number of confirmed cases had risen to 21,209, with 896 deaths [13], resulting in a case-fatality rate of 4.22%.

To be a CHV in Kenya, one needs to have been a member of the community where they are serving for a minimum of 5 years prior to their appointment [14]. They should not expect financial remuneration for their services from the government. They should be "willing and ready to provide services to the community without charging" the community members for the same services [14]. This was the general situation at the time of the study, but since October 2023, they receive a stipend for their services from the County Governments of Kenya (GOK).

CHVs in Kenya typically serve approximately 100–500 individuals depending on the county's population density [15].

## Sampling

To determine the number of CHVs in Kiambu County eligible for selection, the study examined the functional Community Health Units (CHUs) within the county from 2017 to 2020, as shown in Table 1 below.

To participate in the FGDs, CHVs were required to meet the inclusion criterion below.

*Inclusion criteria:* To be eligible for participation in the FGD, a CHV had to have at least 5 years of experience, including the three years within the study period (2017–2020), have been actively involved in the pandemic response, and be available for the interview.

*Exclusion criteria:* Any CHV who did not meet the above stipulated qualifications was excluded.

However, identifying eligible CHVs was challenging due to the consistently low number of functional CHUs prior to 2020 (Table 1) alongside strict pandemic measures. The substantial increase in functional CHUs observed in 2020 reflects the increased demand for community health services driven by the COVID-19 pandemic and the GOK community health strategy during this period. After consultations with CHSCs, it was determined that only 50 CHVs met the inclusion criteria.

Participants were selected through convenience sampling. The pool of eligible participants restricted to 50 CHVs, ensuring a basis for comparison of experiences before and during the pandemic. Therefore, two FGDs were conducted: one comprising 12 women and the other comprising 12 men, resulting in a sample size that represented 48% of the eligible CHVs.

To ensure adequate representation of Kiambu County, which consists of 12 sub-counties, two CHVs were selected from each sub-county. This included 12 participants from upper Kiambu (a predominantly rural area), and 12 participants from lower Kiambu (which is more urbanized).

## Focus group guide development

A literature review was conducted to explore the known impact of the COVID-19 pandemic on the CHVs to direct the content of the FGDs using four key papers [16–19].

Under the collaborative guidance of a sociologist affiliated with Mount Kenya University and MMK, a public health expert representing the County Government of Kiambu, an FGD guide was tailored to the objectives of the present study. This guide was developed in English.

The focus of the FGD was on the experiences of CHVs in performing their roles and responsibilities, particularly in light of the additional duties they were assigned during the pandemic. We aimed to explore whether they were able to effectively carry out their responsibilities under these new circumstances. Given that CHVs regularly interact closely with community members in their homes, we also sought to identify the strategies they employed to overcome the challenges they faced during the lockdown. Furthermore, due to their direct contact with the community, CHVs served as a proxy for understanding community perspectives on COVID-19, including views on its origins, spread, and preventive measures. Finally, recognizing that CHVs represent the first level of healthcare in Kenya (one that should have the broadest coverage), we inquired about their views on the government's handling of the pandemic.

**Table 1. The total number of functional CHUs within Kiambu County during the period under study. The data was derived from the Kenya Health Information System (KHIS).**

| Year | Jan | Feb | Mar | Apr | May | Jun | Jul | Aug | Sept | Oct | Nov | Dec |
|------|-----|-----|-----|-----|-----|-----|-----|-----|------|-----|-----|-----|
| 2017 | 8 | 3 | 3 | 7 | 7 | 3 | 3 | 8 | 1 | 2 | 0 | 0 |
| 2018 | 0 | 3 | 0 | 0 | 6 | 0 | 0 | 0 | 0 | 0 | 0 | 3 |
| 2019 | 1 | 7 | 12 | 19 | 17 | 26 | 8 | 6 | 11 | 11 | 12 | 19 |
| 2020 | 19 | 15 | 1 | 6 | 6 | 6 | 10 | 31 | 77 | 0 | 0 | 0 |

NW and BK then received training on qualitative data collection and conducted the FGDs. This content was collated in the form of an interview guide with six key questions targeting the public health measures for use in the focus groups:

1. Please share your experiences during the COVID lockdown. What challenges did you face, and how did you overcome them?

2. How did the community respond to the COVID-19 control measures?

3. How did the COVID-19 control measures impact community access to health facilities?

4. How did the COVID-19 control measures affect the community's ability to manage other priority diseases?

5. What are the community's views on COVID-19 regarding its origin, spread, and prevention?

6. Do you feel the county was prepared for the COVID pandemic, and is it prepared for another pandemic if it were to occur?

In addition to the development of the FGDs, a desk-based analysis was conducted, the details of which are available in the S2 File.

## Data collection

All participants were given information about the study. All FGD interviews began after receiving signed written informed consent forms from each participant during recruitment. The participants were free to leave at any time during the FGD and this was communicated in the consent form.
In total, two FDGs were conducted on the 4th and 6th June 2021.

The responses to the FGD questions were allowed in either Kiswahili or English, depending on the participants' preference, and the discussions were conducted in a private area of a convenient health facility to ensure free and uninterrupted conversation.

For anonymity, participants did not share their names or any personal identifiers. Instead, they were given numbers for transcription purposes.

All the FGDs were audio recorded. The duration of the FGDs ranged between 1–2 hours, followed by a debrief session.

## Data management and analysis

The Principal Investigator (PI), MMK, served as the County Head of Clinical Research. NW, who conducted the data collection, was the Monitoring and Evaluation Coordinator for Kiambu County's Health Department. The transcriber, PKJ, has extensive experience in transcribing qualitative data.

Data from the audiotapes were transcribed verbatim and translated into English for analysis. Both inductive and deductive data analysis approaches were employed.

Initial transcription was performed by PKJ, with both PKJ and MMK independently reviewing the transcripts and generating initial codes. To ensure rigor and validity, the two researchers (PKJ and MMK) collaboratively reviewed the data during a subsequent meeting, further refining the codes through indexing and charting, and reached a consensus.

The finalized codes were used to categorize segments of the transcripts using MAXQDA 2020 (VERBI Software, Consult. Sozialforschung. GmbH, Berlin 2019), a qualitative data analysis software that facilitates systematic coding, categorization, and organization of textual data. The software also enabled quantification of code frequency, aiding in the identification of patterns within the dataset.

Finally, any discrepancies in the coding process were addressed through discussions among MMK, NW (who also generated their own independent codes), and PKJ, leading to an agreement on the emergent themes.

## Ethical considerations

Ethical approval was sought and granted from The University of Eastern Africa Baraton Ethics Committee, approval number UEAB/REC/08/06/2020. Permission was also obtained from the County Health Department.

This work has been carried out in accordance with The Code of Ethics of the World Medical Association (Declaration of Helsinki) and aimed for the inclusion of representative human populations.

Informed consent was obtained from each participant and no private identifying information was collected before, during or after the study, and the study was voluntary, allowing participants to opt out at any time.

## Results

### Participants

There were 24 participants in the FGDs. The male-to-female ratio was 1:1. The range of experience of the participants was 5–24 years.

### Findings

Part A of this study presents the experiences of the CHVs during the COVID-19 lockdown. Part B presents the perspectives of the community regarding COVID-19 as reported by the CHVs. Part C covers the opinions of the CHVs on the county government's preparedness for COVID as well as a future pandemic.

### Part A: The experiences of the CHVs during the COVID-19 pandemic

**Dedication and commitment to serve the community.** The CHVs demonstrated a strong commitment to service, often prioritizing the needs of the community over their personal fears about COVID-19. They described their efforts to connect patients with healthcare facilities. In some instances, CHVs personally covered the costs of treatment or transportation to ensure clients received the care they needed. Additionally, they participated in training programs aimed at enhancing their knowledge and skills related to COVID-19, enabling them to better support their communities.

*"We did have some challenges such as the diabetics not having any medication but since they knew which medications they were taking, we took the responsibility to go to the hospitals and get a two week or a one-month supply of their medication and give it to them at their house because they were not going to the clinics/hospitals."* **(Resp.1)**

*"People of ANC, the mothers… because I remember there was a place I went for a home visit, the mother told me she hasn't been to the clinic in seven months. Seven months! And she hasn't gone to the clinic. I told her 'Let's go, because these months you don't know the state of your child'. [I convinced her to leave] her house and we went. I only left her once she was seen at the facility, I told her 'Let's not be afraid, let's go. See even me I will be there, let's go'. Then those women who gave birth, they were maybe in the third month and during the 4th month COVID was announced, so many skipped immunizations. They skipped period because when they were told 'stay-at-home. Lockdown.' They didn't have an otherwise."* **(Resp.7)**

*"I serve I was able to overcome my challenges because first of all I was filled with love for the people I serve. When I would visit them, I would ensure that they would wash their hands many times, wear their masks and also that they had sanitizers."* **(Resp.2)**

Despite the acknowledged feelings of demoralization regarding the community's failure to comply to COVID-19 procedures in spite of numerous sensitization exercises, the commitment to their duties was evident.

*"when the government said wear masks, people did not want to wear masks. When they were told to keep distance, you could see that they would stand close to each other. We were very disturbed you felt defeated when you would tell them to keep distance but when you reach there, they are close to each other." (Resp.6)*

Demoralization was compounded by the perception that the government had abandoned them. Basic PPE was inadequately funded, and their expanded responsibilities were insufficiently recognized. Furthermore, their perspectives highlighted the uncertainty they faced regarding infection prevention while performing their duties.

*"We are the ground… what is it called? Zero level. We are the ones who are on the ground. We have been trying to explain that CHVs are people who volunteer. Which is the truth. But us CHVs there is no one who has remembered us… For example, the county itself has not recognized the work us CHVs have done." (Resp.4)*

*"… until we had a program that would assist us with those materials, those masks and sanitizers, which enabled us. But otherwise, you would just go with that fear, even the client would be scared of you." (Resp.2)*

*"…we didn't get any gloves we didn't get any masks we didn't get any anything. You would really apply yourself to your job." (Resp.3)*

These challenges were alleviated through support from non-governmental organizations, religious institutions, and other partners, who supplied CHVs with essential PPE. However, for some organizations, this critical assistance was delayed for several months, leaving CHVs without necessary resources during the initial stages of the pandemic.

*"as a community health volunteer, the way I overcame my challenges was that there was an organization in the sub-county called Living Goods. It has helped us so much as community HWs in Thika sub-county. They used to supply us with soap every month, masks, cotton wool, they would give us even spirit so that we could go to the households and continue on with our work. For example, we would not lack masks. We were given boxes of 50 masks, they gave us gloves, so if I went to the community to do my work at least I was protected." (Resp.4)*

**Barriers to community health service delivery**

The COVID-19 pandemic disrupted the ability of CHVs to perform their duties due to the implementation of public health measures aimed at curbing the spread of the virus. One of the key strategies employed by the government was the restriction of physical movement through the establishment of zonal roadblocks on major public routes. These restrictions had far-reaching consequences, not only limiting patients' access to healthcare facilities for regular appointments but also restricting CHVs' ability to reach their clients. This was particularly problematic for individuals with chronic conditions, such as those attending comprehensive care clinics (CCC) for HIV treatment.

*"As a community volunteer … we didn't go door-to-door to check for the clients that we had. We didn't have equipment that we should have used for that period. And still, the community that needed us didn't have any gathering [due to social distancing] to educate them in various ways." (Resp.11)*

*"…like those patients who would come to CCC, many were far away and there were roadblocks. For instance, maybe they're from Lari and they come to the facility in Tigoni, and now there was a road block at Mathore. So now will they really pass through?" (Resp.12)*

*"okay to add to that I can say that most of the community members found that since there was cessation of movement most transport or public means were not available to them. So, it was required to take a taxi or a motorbike and then also the fare was hiked. Then you get to the roadblock and you asked where you are going and you remove a paper from the hospital and say I am going to hospital. So, to avoid all that you just stay at home." **(Resp.7)***

In addition to the movement restrictions and roadblocks, a curfew was imposed, further restricting people's ability to seek emergency care outside of designated hours. As a result, even though alternative channels for emergency medical services were put in place, these measures were not widely known or accessible. This compounded the challenges of timely response to emergencies, as individuals were unable to receive prompt care during the restricted hours.

*"Because curfew was enforced, even motorbikes didn't go anywhere, cars were not on the road. To have an illness meant you had a lot of problems and not all of the people knew that 1196 number. Many people died because they didn't get access to doctors. they just died." **(Resp.3)***

When CHVs were able to visit patients at their homes, they faced additional challenges in providing care due to the need to observe social distancing and adhere to the government's 'low touch' guidelines. This was further compounded by the lack of appropriate personal protective equipment (PPE) and essential Point of Care Diagnostic devices.

*"You had to somehow keep distance while measuring that child's temperature, and then you had to sanitize everything you used. Then the parents also had to sanitize. it was even better if you showed the mother how to check the temperature and they did it for you. Then after the mother checked the temperature you had to sanitize it again. It was hard work for parents to understand what you are saying, you see you are not close to them…" **(Resp.8)***

**Part B: Community perspectives on and responses to the COVID-19 measures**

**Generalized anxiety.** The CHVs reported that the community members had a generalized anxiety of contracting COVID-19, which caused them to avoid medical institutions. Poor health-seeking behavior ended in reported morbidity and mortality.

*"a patient of mine even had two of her children die, some of them are diabetics and some of them have high blood pressures and they would completely refuse [to go to hospital] they would say 'we will go there to die'." **(Resp.2)***

*".. it really affected the health facilities because people weren't coming to the clinics, they didn't pick up their medicines, they were scared especially if they saw the thermogun, if their temperature was checked they didn't want it to be checked." **(Resp.1)***

The respondents reported that healthcare workers in the facilities, CHVs, sick community members, and foreigners faced stigmatization.

*"Like, most of the community members stigmatize the community health volunteers because most of the health workers had been infected with the disease so we when we went to the communities, we were told that 'you work in the hospital' and 'you work with health workers so you can get COVID and bring it to the rest of us." **(Resp.3)***

**Economic hardship**

According to the CHVs' opinions, initial adherence to COVID-19 requirements gradually decreased as a result of financial consequences. The general perception among the CHVs was that the government had neglected its sick citizens

and failed to provide financial support during the pandemic. Additionally, gender-based violence was seen as a consequence of these financial challenges. There was also a belief that political leaders themselves did not adhere to the health protocols.

> "...there was no money to buy food, there were no jobs, GBV increased again, gender-based violence. Because the husband didn't have money and the wife didn't have money, so you would just hear people were kicked out of their homes, people were fighting each other..." **(Resp.5)**

> "... And it was expected that the government would provide their basic needs but then you aren't at work the kiosk isn't open and, in the evening, you have to eat something. So quite a number of them began to stop adhering to the stay-at-home orders." **(Resp.9)**

> "it reached a point politicians resumed political rallies so people, the community were relying on the leaders by watching how they operate. So, people saw the political rallies are ongoing, so they stopped adhering to the measures. People began to go to meetings. Like the meetings for dowry, those resumed." **(Resp.10)**

**Misinformation about COVID-19**

The community members generally doubted information about COVID-19. It was a prevalent theme that the official narrative about the origins of COVID-19 was not entirely dependable, that the disease did not exist, or that it was a money-making scheme. In addition, according to the community, rural dwelling Kenyans were protected from contracting COVID-19 becauseit was a disease for the rich, foreigners, travellers, or urbanites.

> "Even right now many people say that there is no such thing as COVID." **(Resp.3)**

> "Another side to it is that, the time it came here they believed it was a way for the government to make money. So, they didn't care. Maybe only because of the police they complied with some of the measures." **(Resp.12)**

> ".. there was no person who worked like us CHVs did explaining to people about COVID. They would say this is a government project to look for money from us…" **(Resp.1)**

> "about the community perspective about the origin of COVID, many said that it was a disease from China for Chinese and it can't kill Kenyans…" **(Resp.3)**

They were also misinformed about the mode of transmission of the disease.

> "In the bars, they would say they would say that you are drinking alcohol but you are also washing your hands with sanitizer, so you won't get COVID." **(Resp.10)**

**Part C: CHVs perspective on county governmental readiness for the COVID-19 pandemic and future pandemics**

**County unpreparedness in managing COVID-19.** The majority of the respondents did not believe the county was in a position to handle COVID-19 or future pandemics. This was due to the supposed decay of governmental trust, lack of governmental support in health care, and the disorganized handling of the COVID-19 centre in the county.

> "When we would talk about COVID-19, you would see in the media that NHIF would not assist those who were infected with COVID. It was like there was some sort of discrimination going on. Everyone had to chip in for the funds, but when it came time to use the money for the people who were affected, they could not give them the money. The other day,

*people were asking me, listen, we put this money in NHIF, why can't it be used to help someone who has COVID? What is it for then?" **(Resp.6)***

*"And when we closed Tigoni, you know we closed all the departments? And we left and we went to Limuru Health Centre. That facility did not have the requirements like Tigoni... There is no ward to admit mothers there. There is no ward to admit male or female or paediatric, there is none. Now, they should have prepared first, saying we will close Tigoni first, then we go to a hospital which has things like X-ray, triage, ANC, maternity… but now they just suddenly closed it. We were just forgotten and confused. Because even the community was not prepared. They were not prepared at all." **(Resp.5)***

## Discussion

This study sought to illuminate the experiences, concerns and assessment of government support of the CHVs during the COVID-19 pandemic in Kiambu county, Kenya. In general, the study highlighted that the COVID-19 pandemic was a strain to community health services in Kiambu county. Previous studies (unrelated to pandemics) on community health workers have cited a lack of resources to efficiently carry out their duties [20], high workloads [2], and inadequate transport provisions as some of the key challenges experienced at this level [21], which have been reiterated by this study. In addition to these obstacles, we highlight additional specific challenges and their possible remedies in the following sections.

### CHV dedication

We found that the CHVs demonstrated unwavering perseverance to fulfill their duties in spite of their various constraints. From the emerging themes, this motivation arose from their intrinsic and altruistic desire to serve their community. This is similar to a study conducted in the United Kingdom, though this study collected anonymous data from front-line HWs who worked in hospitals [22]. Another comparative, qualitative study conducted in both Sierra Leone and Nepal reported individual determination, a sense of responsibility to the community and professional duty that compelled staff to stay or return to their workplace [23]. This corroborates the present study results and underpins the resilience possessed by HWs across all tiers of the health system.

However, our findings contrast with studies conducted in other settings, where HWs faced challenges that sometimes led to disengagement or burnout. A study conducted in Kenya highlighted that inadequate supportive supervision and insufficient resources and supplies were significant contributors to the demotivation of CHVs, ultimately leading to poor performance [24]. This phenomenon was not confined to LMIC settings. Similarly, a mixed-methods study conducted in Massachusetts reported that 37% of HWs in the study expressed intentions to leave the profession due to the mental health impacts and burnout due to the COVID-19 pandemic [25].

The dedication displayed by CHVs in this study could be the strong community foundation in rural and peri-urban settings in Kenya. CHVs often live in the same communities as their clients, fostering personal relationships that go beyond professional duties. This cultural connection likely contributed to the CHVs' sense of responsibility to continue serving their communities. This intrinsic motivation has also been described in neighboring Tanzania [26]. Furthermore, support from government partners and NGOs in providing resources helped mitigate some of the challenges faced by CHVs and provided support for their duties.

### Stigma was a barrier

The CHVs experiencing stigma from the community members. This is in tandem with the findings from studies conducted in Guinea, Liberia and Sierra Leone during the Ebola outbreak [27]. There were also similar findings from Uganda during the Marburg and Ebola outbreaks where HWs were ostracized by both the society and family members [21]. The reported

effect in the latter study were so severe, one community HW had to defend himself on radio after recovering from Ebola. Although our study's experiences did not go to that extreme, they do underscore the need for more government and public health professionals to concentrate on health advocacy and sensitization. In contrast to these findings, a study in Sierra Leone found that HWs felt supported by their communities [28]. This support was attributed to the recognition of the HWs' essential roles and the collective effort to combat the outbreak.

### Rampant misinformation within the community

The second-person accounts of the community perceptions of COVID-19 relayed fear of health facilities. A lot of these fears coexisted with misinformed beliefs about the pandemic. This is in line with a study conducted in the United States and Canada where respondents believed HWs should be isolated from their communities and families [29]. For fear of infection, more than a third of respondents avoided HWs in the latter study [29]. Similar to our study, the decay in adherence to COVID-19 public health protocols has also been described previously [30]. This necessitates increased efforts from stakeholders to address this issue in future pandemics, as it has significant implications for both mental health and adherence to health protocols [30].

### Economic strife

The CHVs also highlighted the negative economic effects of the response to COVID-19 on the community members. This is corroborated by another study conducted in Poland where people relayed similar financial constraints as a result of the pandemic measures [31]. A similar impact was reported in a 2021 survey by the Ministry of Health in Kenya [32]. The survey revealed that among respondents who skipped meals, the percentage attributing this to the pandemic increased significantly, from 37% to 98% [32]. There was also a significant increase in the partial loss of income due to the pandemic measures in the latter report. These economic hardships within the community intensified feelings of abandonment by the government and contributed to a decline in adherence to COVID-19 protocols, as individuals struggled to meet basic needs, let alone afford masks and other PPE items. The need for targeted economic recovery efforts is pertinent and this finding highlights the importance of addressing the economic impacts in future pandemics to ensure compliance with public health measures and support for vulnerable populations.

### Perceived shortcomings of the government

The respondents expressed reservations about the county government's handling of COVID-19 and any potential future pandemics. This sentiment was also shared by nurses and doctors in Bangladesh during the COVID-19 pandemic and arose due to broken government promises [33]. In the Bangladesh study, the HWs were disregarded once they contracted COVID-19 and had to settle their hospital bills themselves despite the government promising to do the same. In the current study, respondents highlighted the county government's failure to fulfill promises related to the provision of equipment, psychosocial support, and logistical resources, as outlined in the Government of Kenya's Community Health Guidelines issued by the Ministry of Health [10]. This perceived neglect eroded trust and created doubts about the government's capacity to manage pandemics effectively.

Contrasting this finding is a study from Singapore, where members of the public demonstrated high levels of trust in the government's ability to manage the pandemic. This trust was attributed to the government's transparency and open communication with the public [34]. As COVID-19 began to unfold in Singapore, key government officials in Singapore openly addressed the scientific uncertainties surrounding the virus in plain language. Furthermore, extensive contact tracing procedures, coupled with clear communication from credible sources, created a positive perception of the government's risk management and established a high level of trust in its actions [34].

The contrasting sentiments between Kenya and Singapore, therefore, can be attributed to structural and logistical challenges that often hinder the ability of governments in LMIC settings to meet their commitments during health crises. Issues such as inadequate funding, supply chain difficulties, and competing priorities further exacerbate these challenges. Additionally, in our study, the perception of hypocrisy from government officials and instances of poor role modelling contributed significantly to the mistrust observed in the Kenyan context.

## Limitations

A key limitation of this study is that only two FGDs were conducted; therefore, the findings and conclusions should be interpreted with caution, given the limited extent to which data saturation was achieved.
The limited sample size was primarily due to the availability of only 48% of eligible CHVs during the COVID-19 crisis, as pandemic-related restrictions made recruitment particularly challenging. Although participants were drawn from diverse settings within Kiambu County, the small number of FGDs limits the extent to which these findings can be extrapolated to the broader healthcare landscape in Kenya.

Nonetheless, this study provides a preliminary foundation for future research, particularly given the documented and longstanding systemic neglect of CHVs. Across counties, Level 1 services have been largely overlooked, CHVs have experienced high attrition rates and demoralization, stipends have not been consistently provided, and the policy framework for community health services remains in the process of institutionalization [35–37].

While we acknowledge the study's limitations, the insights presented here contribute to a deeper understanding of the realities faced by CHVs and underscore the need for further research and policy attention.

## Recommendations

Our findings reiterate well-documented barriers to achieving UHC at the community level in Kenya, particularly the challenges of stigma, misinformation, resource constraints, and public trust in government policies [38].

To address these challenges, the GOK should strengthen its public health communication strategies. Effective and culturally sensitive campaigns are essential to counter misinformation and reduce stigma, particularly during health crises. These campaigns should integrate clear, science-based messaging with community engagement efforts to build trust and enhance public adherence to health policies.

Given the pivotal role of CHVs in combating misinformation, ongoing training should emphasize empathy-based approaches to improve their ability to engage with community members.

Strengthening the community health strategy requires increasing both financial and emotional support for CHVs. Adequate compensation, access to mental health resources, and professional development opportunities are essential to reducing attrition and improving recruitment. The recognition of CHVs as community health promoters under the Primary Health Care Act, 2023 [7] is a significant milestone, but sustained investment in their welfare and training remains necessary. These interventions can serve to empower CHVs (now known as "Community Health Promoters"), reinforcing their role as frontline healthcare providers and enhancing the resilience of the healthcare system at the grassroots level.

Beyond CHV support, Kenya must address the persistent funding gaps within its healthcare system to strengthen infrastructure and human resources [39]. Increased investment is needed not only at the community level but also in expanding and equipping higher-tier health facilities that receive patient referrals. Current health budget allocations remain below the Abuja Declaration target, highlighting the urgent need for the government to prioritize healthcare in future fiscal planning [40].

Equipping CHVs with modern tools, such as the Electronic Community Health Information System (eCHIS), can improve community-level data collection, enhance service delivery, support monitoring and evaluation, and promote better integration into the national healthcare system.

Future research should aim to address the study's limitations by employing broader methodologies across multiple counties to enable more generalizable and robust conclusions. A comparative study involving diverse regions and larger sample sizes would provide deeper insights into CHV experiences and inform more targeted policy interventions.

## Conclusion

While the CHVs in Kiambu County embraced their role in the fight against the COVID-19 pandemic, they perceived a lack of support from both the government and the community. This perspective is linked to their stigmatization, experiences of resource constraints that hindered their ability to safely and effectively carry out additional duties during COVID-19, inadequate recognition, and a perceived sense of abandonment. The GOK amended the Primary Health Care Act in 2023, placing CHVs (now known as CHPs) at the forefront of the goal of achieveing UHC and renewed its commitment to community-level healthcare service delivery. This is progress in terms of policy support, but continued efforts are needed to ensure that the necessary resources for the full execution of this strategy are availed. However, given the limitations of this study, future research should build upon these findings to provide a more comprehensive understanding of CHVs' experiences and inform stronger policy interventions.

## Supporting information

**S1 File. Collected data for this study.** This document contains both the audio transcriptions of the focus group discussions translated into English.
(PDF)

**S2 File. Level 1 desk review results.** Contains the desk review on the state of the community health level.
(PDF)

## Acknowledgments

We appreciate the community health strategy fraternity taking the time to respond to all of our inquiries. The study's inception was backed by the County Government of Kiambu and the County Health Management Team, which also granted permission for its members to take part in different roles. Without the support of the Transforming Health Systems for Universal Care (THS_UC) Project, which supported this study's research, this study wouldn't have been possible. There are no tools, instruments, or materials that are protected by copyright in this original work. The authors of this work are responsible for its findings.

## Author contributions

**Conceptualization:** Naomi Wachira, Magoma Mwancha-Kwasa.

**Data curation:** Naomi Wachira, Prabhjot Kaur Juttla, Bernard Kimani, Moses Kamita, Samuel Mungai, Francis Makokha, Magoma Mwancha-Kwasa.

**Formal analysis:** Naomi Wachira, Prabhjot Kaur Juttla, Bernard Kimani, Moses Kamita, Samuel Mungai, James Ndimbii, Francis Makokha, Magoma Mwancha-Kwasa.

**Funding acquisition:** Moses Kamita, Samuel Mungai, Magoma Mwancha-Kwasa.

**Investigation:** Prabhjot Kaur Juttla, Bernard Kimani, Samuel Mungai, James Ndimbii, Magoma Mwancha-Kwasa.

**Methodology:** Naomi Wachira, Prabhjot Kaur Juttla, Bernard Kimani, Samuel Mungai, James Ndimbii, Magoma Mwancha-Kwasa.

**Project administration:** Naomi Wachira, Bernard Kimani, Samuel Mungai, James Ndimbii, Magoma Mwancha-Kwasa.

**Resources:** Naomi Wachira, Bernard Kimani, James Ndimbii, Magoma Mwancha-Kwasa.

**Supervision:** Naomi Wachira, Francis Makokha, Magoma Mwancha-Kwasa.

**Validation:** Naomi Wachira, Prabhjot Kaur Juttla, Moses Kamita, James Ndimbii, Francis Makokha, Magoma Mwancha-Kwasa.

**Writing – original draft:** Naomi Wachira, Prabhjot Kaur Juttla, Bernard Kimani, Moses Kamita, Samuel Mungai, James Ndimbii, Francis Makokha, Magoma Mwancha-Kwasa.

**Writing – review & editing:** Naomi Wachira, Prabhjot Kaur Juttla, Bernard Kimani, Moses Kamita, Samuel Mungai, James Ndimbii, Francis Makokha, Magoma Mwancha-Kwasa.

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
