## [Decision Letter · Decision Letter 0]

26 Nov 2024

PONE-D-24-48602Community health volunteers’ experiences during the COVID-19 pandemic in Kiambu county, Kenya: a qualitative studyPLOS ONE

Dear Dr. Juttla,

Thank you for submitting your manuscript to PLOS ONE. After careful consideration, we feel that it has merit but does not fully meet PLOS ONE’s publication criteria as it currently stands. Therefore, we invite you to submit a revised version of the manuscript that addresses the points raised during the review process.

We look forward to receiving your revised manuscript.

Kind regards,

Olushayo Oluseun Olu

Academic Editor

PLOS ONE

Journal Requirements:

Reviewers' comments:

Reviewer's Responses to Questions

**Comments to the Author**

1. Is the manuscript technically sound, and do the data support the conclusions?

Reviewer #1: Partly

Reviewer #2: Yes

2. Has the statistical analysis been performed appropriately and rigorously?

Reviewer #1: N/A

Reviewer #2: Yes

3. Have the authors made all data underlying the findings in their manuscript fully available?

Reviewer #1: Yes

Reviewer #2: Yes

4. Is the manuscript presented in an intelligible fashion and written in standard English?

Reviewer #1: No

Reviewer #2: Yes

5. Review Comments to the Author

Reviewer #1: Abstract

1. The conclusion was poorly written and should be revised.

Main Manuscript

2. The introduction aspect of the paper needs to be reorganized to ensure flow and sequence. In addition, the authors were not convincing in their justification or rationale for this study.

3. Which kind of support was provided to the CHVs prior to the promulgation of law that recognized their role in the health system?

4. How many health facilities are this county, and what kind of services do they provide, especially during the COVID-19 pandemic?

5. What is/are the roles of the CHVs (not CHPs) played in relation to these HFs or various levels of care, if any?

6. I suggest the study population and sampling should come after the study design.

7. What do you mean by this: “The data was collected as part of a larger quasi-experimental mixed methods pre-post study”?

8. The selection of CHVs was somewhat opaque and unspecific. Can you please elaborate on the selection of CHVs for the FGD sessions?

9. How was the desired number of participants determined?

10. Line 139, why do you have to use the word exigencies? I suggest you use the word needs or requirements

11. Were the participants informed of their preserved right to quit the study at any time? Was it verbal or in writing?

12. The PI (MMK) was the County Head of Clinical Research while the researcher (NW) who did data collection NW was the Monitoring and Evaluation Coordinator. Can you please re-write this?

13. There seem to be contradictions in the emerging ideas from the FGDs considering the barriers and their roles.

14. At what point in the COVID-19 pandemic were the CHVs provided with kits, such as face masks, by the NGO- Living Goods, given that a major limitation to service delivery by the CHVs was the lack of appropriate PPE and basic devices at the point of care?

15. This is in keeping with studies conducted in Guinea, Liberia and Sierra Leone during the Ebola outbreak (23). How many studies have you cited here? Kindly correct this.

16. “The present study also highlighted the negative economic effects of the response to COVID-19 on the community members. This is corroborated by another study conducted in Poland where people relayed similar financial destitution as a result of the pandemic measures (25).” This does not tally with the sub-topic of misinformation or being scared about the disease. Perhaps, this (economic implication) can be discussed elsewhere.

17. Regarding disappointment in government, the authors should discuss what they found (results) in the current study, then compare it with other studies (local and international) and proffer possible explanations, where relevant.

18. The discussion requires a lot of improvement as it seems shallow and unconvincing.

19. There are other real and potential limitations and biases in the study that should have been stated in the manuscript. The authors need to think through this.

20. The entire manuscript requires editing.

21. On the recommendation, this study was conducted before the CHVs (community health strategy) were formally integrated into and compensated by the formal health system by the GoK. Hence, how does this study adequately identify the challenges and obstacles towards realizing the UHC goal in the context of the roles of the CHVs given the temporal change(s)?

22. The recommendations are lengthy and somewhat boring. The authors should summarise and make this section concise.

23. The discussion, conclusion and recommendation require considerable revision with a clear flow of thought and reorganization, in alignment with the objectives of the study.

Reviewer #2: Community health volunteers’ experiences during the COVID-19 pandemic in Kiambu county, Kenya: a qualitative study

GENERAL COMMENT.

This is generally a good research manuscript. The choice of the study population is appropriate, and the context of investigation is also very applicable. There are documented facts that COVID-19 pandemic revealed the weaknesses in the health systems across WHO African countries and most especially its implications regarding the human resources for health (HRH) is palpable. The inclusion of the CHVs in Kenya was supposed to be helpful in taming the scourge, but this research is an eye opener to the challenges associated with their duties and the gaps in governance in general at the study location. The comments below should be addressed by authors and if this is done dutifully, the manuscript can be accepted by the journal for publication.

SECTIONAL COMMENTS

Line 22-23: You may want to use another suitable word here…. like ‘CHVs participated in responding to the crisis’…….

Line 24-25: Expunge this phrase ‘Given that the test of a healthcare system is a health crisis….it is not necessary

Line 30 -31: A county in Kenya is a big geographical area with sizeable population covered by appreciable number of CHVs. Please clarify the reason behind limiting the FGDs to only 2. Ensuring saturation of information in any qualitative study is very key to avoid bias in findings

Line 40: The word ‘disappointed’ should be replaced with another suitable word. It looks like the researcher was emotionally drawing a conclusion and may not necessarily represent the findings from the study

Line 94: On a general note, data collection in qualitative study should harness multiple sources of information, including Focused Group Discussion (FGD), Key Informant Interview (KII) and possibly document review. This will facilitate information richness and reduce bias to the barest minimum. However, if only FGD was to be used, it was expected that it would be done with multiple groups with adoption of saturation principle. Another acceptable option is to conduct interview for reasonable sample (with applicable sampling formular) of the CHVs using self-administered semi-structured questionnaire. I’d advise that the authors consider this suggestion as important.

Line 100: It will be good if the researcher could add other related county statistics as regards the number of health facilities, health workers/population ratio, and possibly the prevalence of the COVID-19 along with case fatality rate in the county as of the study period.

Line 128-129: Previous pandemic or COVID-19? If for the COVID-19, for how long before the time of their selection?

Line 151-152: You may just replace this with…..’Two of the research-authors’. Then how were you able to circumvent bias in the qualitative data collection knowing fully well that two of the authors served as data collectors?

Line 154-164: The researcher should try to summarize what the FGD questions are all about. There is no need to itemize all of them

Line 170-171: Please refer to my earlier comment on this in your abstract. Why limited to 2 FGD only, when saturation level needs to be achieved?

Line 184: Write PI in full.

Line 192: If there was no previous description of MAXQDA2020, the authors may want to add few lines to clarify that

Line 197: Where is The University of Eastern Africa Baraton located? Is it in the same county for study location? If not, why the University?

Line 211: Perspective of the community on what?

Line 22-267: The author should check for full quotation of the respondent’s feedback. I guess the mother has a child with a sickness that was attended to in 7 months. In addition, the phrase ‘I removed her’ should be rephrased as appropriate

Line 231; The quote is too short/. Author should Add a preceding statement to make the quote more comprehensive. Something must have been reported as a positive before the respondent is filled with joy.

Line 315-316: The right way to report should be ….’The respondents reported that there was stigmatization’…..

Line 324: Use ‘According to CHVs’ opinion….

Line 327-328: As this statement ;323 ‘’The political leaders themselves did not adhere to the protocols’’ was the perception of the CHVs, the author should reflect this tone as required. A good option is to rephrase as much as possible

Line 408; Authors can replace with ‘’’This is in tandem with the findings from studies conducted…..’

Line 418-425: This is good. However, I expect to see other side of the arguments where other researchers’ findings contradicts this study’s findings, and they give a balanced few-lines discussion around this. This applicable to other sub-sections of the discussion

Line 427: This phrase sound biased. One county can not stand alone to conclude that the government is disappointing. The author should find another word/phrase to convey the minds of the respondents and should not report with emotions.

Line 428: The author may want to use another word apart from ‘pessimistic’ ...Another word like ‘respondents expressed reservations’…….

Line 444: On a general note, the recommendation section should be summarized by the authors. Maximum of 3 paragraphs should be adequate to summarize the salient recommendations.

Lines 445-447: Authors should not insinuate that the findings are generalisable to the whole country of Kenya. I guess a possible recommendation should be that a possible comparative study using broader method should be done across multiple counties. A better inference can be drawn if this is done in the future.

6. PLOS authors have the option to publish the peer review history of their article (what does this mean? ). If published, this will include your full peer review and any attached files.

**Do you want your identity to be public for this peer review?** For information about this choice, including consent withdrawal, please see our Privacy Policy .

Reviewer #1: No

Reviewer #2: No

---

## [Author Response · Author response to Decision Letter 1]

13 Jan 2025

Editor-in-Chief

PLOS One

9th January 2025

Dear Editor-in-Chief,

I hope this letter finds you well. I am writing to provide our response to the reviewers' comments on our manuscript PONE-D-24-48602 “Community health volunteers’ experiences during the COVID-19 pandemic in Kiambu county, Kenya” submitted to PLOS One.

In response to the reviewers' comments, we have made the relevant revisions to our work. We believe that these revisions address the concerns raised by the reviewers and strengthen the clarity and focus of our manuscript.

We also take this opportunity to thank the reviewers and the editor for your invaluable support and critique that has undoubtedly improved the quality of our work.

Thank you for considering our manuscript for publication in PLOS One. We look forward to hearing from you regarding the next steps in the review process.

Yours sincerely,

Prabhjot Kaur Juttla.

Comments to the Author

Reviewer #1:

Abstract

1. The conclusion was poorly written and should be revised.

Response: We have carefully revised the conclusion for improvement. Please have a look at it and thank you for this observation.

Main Manuscript

2. The introduction aspect of the paper needs to be reorganized to ensure flow and sequence. In addition, the authors were not convincing in their justification or rationale for this study.

Author’s response: Thank you for your feedback. We have completely revised the introduction to improve its structure, ensuring a more logical flow and better sequence. Additionally, we have strengthened the justification and rationale for the study.

Kindly find the new introduction below:

INTRODUCTION

Community health volunteers (CHVs) are recognized by the World Health Organization (WHO) as "a variety of community health auxiliaries who work in their own communities" (1). These volunteers play a critical role in advancing universal health coverage (UHC), particularly in low- and middle-income countries (LMICs) where access to essential healthcare services remains a significant challenge (2,3). In these settings, UHC relies heavily on the extension of primary health care (PHC) into local communities, with CHVs serving as the first point of contact for healthcare delivery and contributing to the effectiveness of healthcare systems (4).

In Kenya, CHVs are key participants in the national community health strategy, which is designed to promote community involvement in healthcare decision-making and delivery. The strategy is managed by community health strategy coordinators (CHSCs), who oversee CHV activities and ensure that volunteers have access to the necessary tools and resources for service provision. CHVs are entrusted with the important task of promoting preventive health measures, supporting health education, and ensuring the continuity of care, especially at the grassroots level (10).

Historically, CHVs in Kenya operated in an informal capacity, without formal recognition or integration into the national healthcare system (11). Before the enactment of the Public Health Act in October 2023, CHVs functioned as volunteers, largely driven by local initiatives and their personal commitment to improving public health (12). Their roles, which primarily involved delivering health information and encouraging healthcare-seeking behaviour, were ad hoc and lacked structure (11). Training for CHVs, when provided, was inconsistent and varied across regions, without standardized guidelines to ensure uniformity in knowledge and skill. Moreover, these volunteers often operated without compensation, relying on their dedication to improving health outcomes in underserved areas.

The outbreak of the COVID-19 pandemic further highlighted the importance of CHVs in the Kenyan healthcare system. Since the first confirmed case in March 2020 (5), the pandemic disrupted healthcare systems globally, and Kenya was no exception. In response, CHVs were enlisted as vital participants in Kenya’s COVID-19 Response Strategy (5,8). Their roles expanded to include promoting behavioral changes, combating misinformation, encouraging hygienic practices, and sustaining efforts to address other priority diseases at the community level (7,8). CHVs also contributed to contact tracing efforts and monitored vulnerabilities within their communities. Despite these critical contributions, CHVs continued to work as volunteers throughout the pandemic, depending on the government, through CHSCs, for the provision of personal protective equipment (PPE), sanitizers, and other necessary supplies (10). They were also expected to receive psychosocial support to help manage the increased demands of their roles. Unfortunately, these measures were often insufficient, underscoring gaps in Kenya’s healthcare system and raising concerns about its preparedness and resilience during health crises.

Given the ongoing legislative reforms aimed at strengthening Kenya’s healthcare system and advancing UHC (9), it is crucial to examine the experiences of CHVs during the pandemic. This study seeks to explore the challenges faced by CHVs and the community members they served while implementing COVID-19 control measures. By analyzing their lived experiences and evaluating their perspectives on government crisis management at the community level, this research offers insights into the role of CHVs and contributes to the broader discourse on improving healthcare system preparedness in LMICs.

3. Which kind of support was provided to the CHVs prior to the promulgation of law that recognized their role in the health system?

Response: Thank you for this observation. We have detailed the same in the introduction section of the manuscript above, and this is a key detail regarding the CHVs past roles and responsibilities and reflects their condition at the time of the study.

4. How many health facilities are this county, and what kind of services do they provide, especially during the COVID-19 pandemic?

Response: we have added this information to the study setting section of our study

“At the time of the study, the county was served by a total of 98 health facilities, distributed across various levels of care as follows: three Tier 5 inter-county facilities, 13 Tier 4 hospitals, 24 Tier 3 health centres, and 70 Tier 2 dispensaries. The number of Level 1 facilities is consistent with the above distribution. During the COVID-19 pandemic, the county adopted a centralized approach to managing COVID-19 cases by designating Tigoni Level 4 Hospital as the exclusive facility for COVID-19 services. Consequently, all COVID-19 patients were referred to this hospital at the time of the study.”

5. What is/are the roles of the CHVs (not CHPs) played in relation to these HFs or various levels of care, if any?

Author’s response: Before 2023, Community Health Volunteers (CHVs) in Kenya played a critical (but rudimentary) role in bridging the gap between communities and healthcare facilities across various levels of care, despite their informal status and limited recognition within the formal health system. They were essential in promoting health and raising basic awareness by educating community members on disease prevention, hygiene, nutrition, and maternal and child health. CHVs encouraged the utilization of health facilities for preventive and curative services, fostering trust and knowledge about available healthcare resources. This has been mentioned in our introduction section.

6. I suggest the study population and sampling should come after the study design.

Author’s response: Thank you for this observation. I have placed the study population and sampling after the study design.

7. What do you mean by this: “The data was collected as part of a larger quasi-experimental mixed methods pre-post study”?

Author’s response: This paper on Community Health Volunteers (CHVs) was part of a broader study examining the impact of COVID-19 on Level 1 healthcare services. The study employed multiple methodologies, including focus group discussions (FGDs), a desk review conducted by the Health Records and Information Officer (HRIO) in Kiambu County, and key informant interviews with members of the County Health Management Team (CHMT). The desk review revealed critical challenges faced by CHV units, including their limited numbers, high attrition rates, and inadequate access to essential commodities.

During the FGDs, findings from the desk review were presented to the CHVs for validation. However, the CHVs strongly disowned the data, expressing confusion and stating that they did not recognize or understand the data reported in the Kenya Health Information System (KHIS). This raised concerns about the reliability of the desk review findings and the potential disconnect between reported data and ground-level realities. We have included this desk review as Supplementary Material File S2. Additionally, while key informant interviews were conducted with CHMT leaders, their responses were inconsistent and varied widely, making it difficult to derive actionable insights from this component of the study. However, we have now expunged this phrase.

8. The selection of CHVs was somewhat opaque and unspecific. Can you please elaborate on the selection of CHVs for the FGD sessions?

Author’s Response: Thank you for raising this important point. To determine the number of CHVs in Kiambu County eligible for selection, the study examined the functional Community Health Units (CHUs) within the county from 2017 to 2020, as shown in the table 1 below, derived from the Kenya Health Information System (KHIS).

SAMPLING

Therefore, to determine the number of CHVs in Kiambu County eligible for selection, the study examined the functional Community Health Units (CHUs) within the county from 2017 to 2020, as shown in Table 1 below.

Table 1: The total number of functional CHUs within Kiambu County during the period under study. The data was derived from the Kenya Health Information System (KHIS).

Year Jan Feb Mar Apr May Jun Jul Aug Sept Oct Nov Dec

2017 8 3 3 7 7 3 3 8 1 2 0 0

2018 0 3 0 0 6 0 0 0 0 0 0 3

2019 1 7 12 19 17 26 8 6 11 11 12 19

2020 19 15 1 6 6 6 10 31 77 0 0 0

To participate in the FGDs, CHVs were required to meet the inclusion criterion below.

Inclusion criteria: To be eligible for participation in the FGD, a CHV had to have at least 5 years of experience, including the three years within the study period (2017–2020), have been actively involved in the pandemic response, and be available for the interview.

Exclusion criteria: Any CHV who did not meet the above stipulated qualifications was excluded.

However, identifying eligible CHVs was challenging due to the consistently low number of functional CHUs prior to 2020 (Table 1). The substantial increase in functional CHUs and the heightened engagement of CHVs observed in 2020 reflects the increased demand for community health services driven by the COVID-19 pandemic and the GOK community health strategy during this period. After consultations with CHSCs, it was determined that only 50 CHVs met the inclusion criteria.

Participants were selected through convenience sampling. The pool of eligible participants was restricted to 50 CHVs who met the inclusion criteria, ensuring a basis for comparison of experiences before and during the pandemic. Two FGDs were conducted: one comprising 12 women and the other comprising 12 men, resulting in a sample size that represented 48% of the eligible CHVs.

To ensure adequate representation of Kiambu County, which consists of 12 sub-counties: two CHVs were selected from each sub-county. This included 12 participants from upper Kiambu, a predominantly rural area, and 12 participants from lower Kiambu, which is more urbanized.

9. How was the desired number of participants determined?

Author’s response: Kindly note our response above.

10. Line 139, why do you have to use the word exigencies? I suggest you use the word needs or requirements

Author’s response: we have used the word “Needs” instead. Thank you for this.

11. Were the participants informed of their preserved right to quit the study at any time? Was it verbal or in writing?

Author’s response: We have clarified the same as below “The participants were free to leave at any time during the FGD and this was communicated in the consent form.”

12. The PI (MMK) was the County Head of Clinical Research while the researcher (NW) who did data collection NW was the Monitoring and Evaluation Coordinator. Can you please re-write this?

Author’s response: We have rewritten this as below for additional clarity: “The Principal Investigator (PI), MMK, served as the County Head of Clinical Research. The researcher, NW, who conducted the data collection, was the Monitoring and Evaluation Coordinator. The transcriber, PJ, has extensive experience in transcribing qualitative data.”

13. There seem to be contradictions in the emerging ideas from the FGDs considering the barriers and their roles.

Author’s response: Thank you for this observation. We have now reworked this section to remove any contradicting ideas and stuck to the physical barriers to performing their duties, as below:

Barriers to community health service delivery

The COVID-19 pandemic disrupted the ability of CHVs to perform their duties due to the implementation of public health measures aimed at curbing the spread of the virus One of the key strategies employed by the government was the restriction of physical movement through the establishment of zonal roadblocks on major public routes. These restrictions had far-reaching consequences, not only limiting patients' access to healthcare facilities for regular appointments but also restricting CHVs’ ability to reach their clients. This was particularly problematic for individuals with chronic conditions, such as those attending comprehensive care clinics (CCC) for HIV treatment.

“As a community volunteer … we didn’t go door-to-door to check for the clients that we had. We didn’t have equipment that we should have used for that period. And still, the community that needed us didn’t have any gathering [due to social distancing] to educate them in various ways.” (Resp.11)

“…like those patients who would come to CCC, many were far away and there were roadblocks. For instance, maybe they’re from Lari and they come to the facility in Tigoni, and now there was a road block at Mathore. So now will they really pass through?” (Resp.12)

“okay to add to that I can say that most of the community members found that since there was cessation of movement most transport or public means were not available to them. So, it was required to take a taxi or a motorbike and then also the fare was hiked. Then you get to the roadblock and you asked where you are going and you remove a paper from the hospital and say I am going to hospital. So, to avoid all that you just stay at home.” (Resp.7)

In addition to the movement restrictions and roadblocks, a curfew was imposed, further restricting people's ability to seek emergency care outside of designated hours. As a result, even though alternative channels for emergency medical services were put in place, these measures were not widely known or accessible. This compounded the challenges of timely response to emergencies, as individuals were unable to receive prompt care during the restricted hours.

“Because curfew was enforced, even motorbikes didn’t go anywhere, cars were not on the road. To have an illness meant you had a lot of problems and not all of the people knew that 1196 number. Many people died because they didn’t get access to doctors. they just died.” (Resp.3)

When CHVs were able to visit patients at their homes, they faced additional challenges in providing care due to the need to observe social distancing and adhere to the government’s ‘low touch’ guidelines. This was further compounded by the lack of appropriate personal protective equipment (PPE) and essential Point of Care Diagnostic (PoCD) devices.

“You had to somehow keep distance while measuring that child’s temperature, and then you had to sanitize everything you used. Then the parents also had to sanitize. it was even better if you showed the mother how to check the temperature and they

---

## [Editor Report · Decision Letter 1]

5 Feb 2025

PONE-D-24-48602R1Community health volunteers’ experiences during the COVID-19 pandemic in Kiambu county, Kenya: a qualitative studyPLOS ONE

Dear Dr. Juttla,

Thank you for submitting your manuscript to PLOS ONE. After careful consideration, we feel that it has merit but does not fully meet PLOS ONE’s publication criteria as it currently stands. Therefore, we invite you to submit a revised version of the manuscript that addresses the points raised during the review process.

**The issue of the conduction of only 2 FGDs vis-a-vis saturation of information is a challenge that you still have to address. I would suggest that you include this in the study limitation section and discuss its impact on the reliability of your findings, conclusions and recommendations.**

We look forward to receiving your revised manuscript.

Kind regards,

Olushayo Oluseun Olu

Academic Editor

PLOS ONE

**Journal Requirements:**

**Additional Editor Comments:**

Thanks for comprehensively addressing all the reviewers' comments. However, the issue of the conduction of only 2 FGDs vis-a-vis saturation of information is a challenge that you still have to address. I would suggest that you include this in the study limitation section and discuss its impact on the reliability of your findings, conclusions and recommendations.

---

## [Author Response · Author response to Decision Letter 2]

17 Mar 2025

Editor-in-Chief

PLOS One

17th March 2025

Dear Editor-in-Chief,

We appreciate the opportunity to revise our manuscript, “Community Health Volunteers’ Experiences During the COVID-19 Pandemic in Kiambu County, Kenya” (PONE-D-24-48602), submitted to PLOS ONE.

In response to the editor’s comments, we have carefully revised the manuscript to address the concerns raised and believe these revisions have strengthened its clarity and focus.

We sincerely thank the reviewers and the editor for their constructive feedback, which has undoubtedly improved the quality of our work.

Thank you for your time and consideration. We look forward to your feedback on the revised submission and the next steps in the review process.

Yours sincerely,

Prabhjot Kaur Juttla.

Editor’s comments

The issue of the conduction of only 2 FGDs vis-a-vis saturation of information is a challenge that you still have to address. I would suggest that you include this in the study limitation section and discuss its impact on the reliability of your findings, conclusions and recommendations.

Author’s response:

Thank you for the opportunity to address this important concern. We acknowledge the limitations associated with conducting only two FGDs and have made every effort to transparently discuss its impact throughout the manuscript. Specifically, we have explicitly addressed this issue in the abstract, study limitations, recommendations, and conclusion sections. Furthermore, in the sampling section, we have provided the raw numbers of the functional units available at the time of the study, which directly constrained our sample size. Recognizing that this limitation affects the reliability and generalizability of our findings, we have carefully framed our conclusions and recommendations accordingly.

Please find the revisions highlighted in bold blue within the relevant sections following this response (below). We hope these clarifications are sufficient, and we appreciate your consideration for publication.

Edited points in the manuscript:

Abstract line 32-27, and line 50-53

Methods: This study employed a phenomenological exploratory qualitative design. Due to logistical constraints, only two focus group discussions could be conducted with 24 CHVs, representing 48% of eligible participants who met the inclusion criteria: a minimum of five years of experience, active involvement in the pandemic response, and availability for the interview. Data were digitally recorded, transcribed, translated, and coded for thematic analysis.

Conclusion: ... However, the study's limitations, particularly the small number of FGDs and lack of data saturation, should be considered when interpreting the findings. Nonetheless, it provides a valuable starting point for further research and comparative analysis across other counties in Kenya.

Limitations, line 540-558

A key limitation of this study is that only two FGDs were conducted; therefore, the findings and conclusions should be interpreted with caution, given the limited extent to which data saturation was achieved. The absence of established saturation affects the reliability of the findings and constrains their broader applicability. The limited sample size was primarily due to the availability of only 48% of eligible CHVs during the COVID-19 crisis, as pandemic-related restrictions made recruitment particularly challenging. Although participants were drawn from diverse settings within Kiambu County, encompassing both lower Kiambu (urban and peri-urban areas) and upper Kiambu (rural areas), the small number of FGDs limits the extent to which these findings can be extrapolated to the broader healthcare landscape. Future research should continue data collection until a valid point of saturation is reached to enhance the depth and robustness of the findings.

Nonetheless, this study provides a preliminary foundation for future research, particularly given the longstanding neglect of CHVs and the systemic challenges they face. Across counties, Level 1 services have been largely overlooked, CHVs have experienced high attrition rates and demoralization, stipends have not been consistently provided, and the policy framework for community health services remains in the process of institutionalization (35–37). While we acknowledge the study's limitations, the insights presented here contribute to a deeper understanding of the realities faced by CHVs and underscore the need for further research and policy attention.

Recommendations, line 561-562 and 589-592

While this study is limited by the lack of data saturation, the insights it provides remain valuable for informing best practices in community health. The findings reiterate well-documented barriers to achieving UHC at the community level in Kenya, particularly the challenges of stigma, misinformation, resource constraints, and public trust in government policies (38).

Future research should aim to address the study’s limitations by employing broader methodologies across multiple counties to enable more generalizable and robust conclusions. A comparative study involving diverse regions and larger sample sizes would provide deeper insights into CHV experiences and inform more targeted policy interventions.

Conclusion, line 603-606

While the CHVs in Kiambu County embraced their role in the fight against the pandemic, they felt that they lacked support from both the government and the community. This perspective is tied to their stigmatization, experience of resource constraints that hindered their ability to safely and effectively carry out additional duties during COVID-19, lack of recognition, and a perceived sense of abandonment. The government has recently renewed its commitment to community-level healthcare service delivery under the guise of “prevention and promotion,” but continued efforts are needed to ensure that the necessary resources for the full execution of the Community Health Strategy are made available. This will be crucial in advancing a community-based approach to achieving UHC. However, given the limitations of this study, particularly the small number of FGDs and the lack of data saturation, future research should build upon these findings to provide a more comprehensive understanding of CHVs' experiences and inform stronger policy interventions.

---

## [Editor Report · Decision Letter 2]

25 Mar 2025

Community health volunteers’ experiences during the COVID-19 pandemic in Kiambu county, Kenya: a qualitative study

PONE-D-24-48602R2

Dear Dr. Juttla,

We’re pleased to inform you that your manuscript has been judged scientifically suitable for publication and will be formally accepted for publication once it meets all outstanding technical requirements.

Kind regards,

Olushayo Oluseun Olu

Academic Editor

PLOS ONE
---

## [Editor Report · Acceptance letter]

PONE-D-24-48602R2

PLOS ONE

Dear Dr. Juttla,

I'm pleased to inform you that your manuscript has been deemed suitable for publication in PLOS ONE. Congratulations! Your manuscript is now being handed over to our production team.

Kind regards,

on behalf of

Dr. Olushayo Oluseun Olu

Academic Editor

PLOS ONE